# Transfer efficiency of organic carbon in marine sediments

**James A. Bradley** [1,2,7] ✉, **Dominik Hülse** [3,4,7], **Douglas E. LaRowe**[5] & **Sandra Arndt**[6]

Quantifying the organic carbon (OC) sink in marine sediments is crucial for assessing how the marine carbon cycle regulates Earth's climate. However, burial efficiency (BE) – the commonly-used metric reporting the percentage of OC deposited on the seafloor that becomes buried (beyond an arbitrary and often unspecified reference depth) – is loosely defined, misleading, and inconsistent. Here, we use a global diagenetic model to highlight orders-of-magnitude differences in sediment ages at fixed sub-seafloor depths (and vice-versa), and vastly different BE's depending on sediment depth or age horizons used to calculate BE. We propose using transfer efficiencies ($T_{eff}$'s) for quantifying sediment OC burial: $T_{eff}$ is numerically equivalent to BE but requires precise specification of spatial or temporal references, and emphasizes that OC degradation continues beyond these horizons. Ultimately, quantifying OC burial with precise sediment-depth and sediment-age-resolved metrics will enable a more consistent and transferable assessment of OC fluxes through the Earth system.

Quantifying feedbacks between the carbon cycle and climate requires knowledge of organic carbon (OC) fluxes between Earth's main reservoirs. The ocean's biological carbon pump (BCP) delivers OC from the sunlit ocean to the deep sea, where it can be buried and sequestered in sediments over geological timescales. Variations in the long-term OC burial rate have played an important role in regulating atmospheric $O_2$ and $CO_2$ over Earth's history[1,2], and potentially contributed to glacial-interglacial cycles[3]. Geologic sequestration of OC relies ultimately on removal of OC from the active carbon cycle by burial in marine sediments and incorporation into the solid Earth. Burial efficiency (BE) is a commonly-used metric to assess the burial versus degradation of OC in marine sediments. It thus serves as an important link in quantifying the flux of OC between fast-cycling surficial reservoirs (i.e., the ocean, atmosphere, biosphere, soils, upper sediments) and geological reservoirs (i.e., deeper sediments, crustal rocks) that cycle slowly over timescales of thousands to millions of years. BE is loosely defined as the percentage of the OC

deposited on the seafloor that becomes buried. Similar to assessing the BCP in the ocean[4], the benthic BE metric requires that a particular reference depth beneath the seafloor ($z_{ref}$) is prescribed, beyond which OC is considered 'buried' and ostensibly 'preserved'. However, OC continues to be degraded beyond these horizons, which are often unspecified. Furthermore, different depth horizons can represent vastly different timescales of burial (largely due to differences in local sedimentation rates). The lack of clearly defined reference horizons for the calculation of BE renders this idealized notion of OC burial and preservation imprecise, inconsistent, misleading, and vague. It thus hinders the comparability of benthic OC fluxes between studies, sites, and reservoirs.

Specifically, BE at a certain depth ($z_{ref}$) beneath the seafloor (hereafter $BE_{depth}$) is the percentage of the OC flux through the sediment-water interface (SWI) ($F_{SWI}$) that is transferred to depth $z_{ref}$ ($F_z$) (Fig. 1). Assuming steady-state conditions (i.e., that the sum of OC degradation (during its transit from the SWI to $z_{ref}$) and burial (i.e., the

[1]Queen Mary University of London, London, UK. [2]GFZ German Research Center for Geosciences, Potsdam, Germany. [3]University of California, Riverside, Riverside, CA, USA. [4]Max-Planck-Institute for Meteorology, Hamburg, Germany. [5]University of Southern California, Los Angeles, CA, USA. [6]Université Libre de Bruxelles, Brussels, Belgium. [7]These authors contributed equally: James A. Bradley, Dominik Hülse. ✉e-mail: jbradley.earth@gmail.com

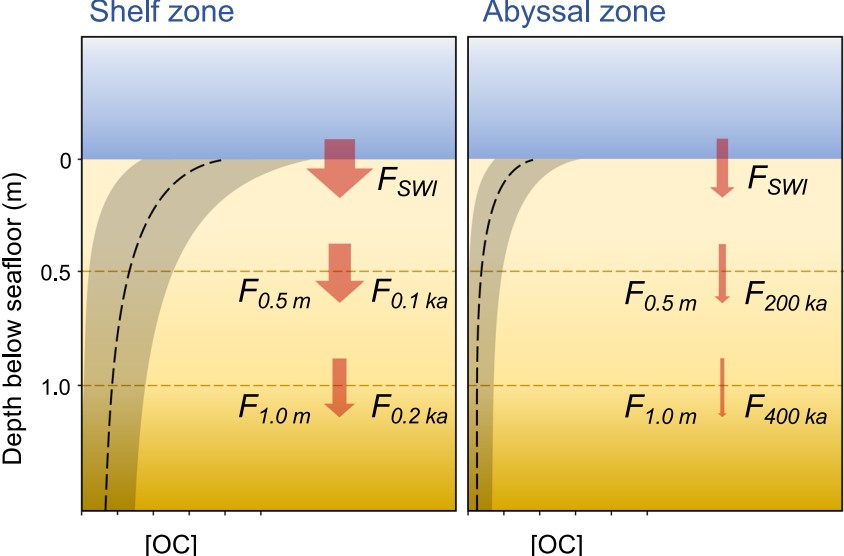

**Fig. 1 | Schematic of the deposition and burial of organic carbon (OC) in idealized marine sediments in shelf and abyssal zones.** The dashed black lines represent illustrative OC concentrations ([OC]) for shelf and abyssal sediments at certain depths and their equivalent (exemplar) ages, and the dark shading represents possible variability in OC concentration between sites. The red arrows indicate the flux of OC through the sediment water interface ($F_{SWI}$), as well as through specific depth or age layers ($F_{depth}$ (e.g., $F_{0.5\,m}$), and $F_{age}$ (e.g., $F_{0.1\,ka}$), respectively).

The widths of the red arrows represent the magnitude of the OC fluxes through those layers. In shelf sediments, OC is rapidly degraded near the sediment-water interface, where shallow sediment depths correspond to short burial times. Conversely, in abyssal sediments, low concentrations of OC persist over long timescales. In the deep ocean, sediments buried at shallow depths (beneath the SWI) have much longer burial times than sediments of an equivalent depth (beneath the SWI) in shallow water. This is due to low sedimentation rates in abyssal zones.

flux of OC through $z_{ref}$) balances the OC flux through the SWI), $BE_{depth}$ is calculated according to:

$$BE_{depth} = \frac{100 \times F_z}{F_{SWI}} \tag{1}$$

The burial depth-horizon ($z_{ref}$) is intended to be the lower limit of the zone within which early diagenesis occurs—which, under steady-state conditions, is represented by the point at which the change in OC concentration ($OC$) with sediment depth ($z$) reaches zero (i.e., $\delta OC/\delta z = 0$)[5].

We note the following issues with the $BE_{depth}$ metric:

I.   OC is never irreversibly 'buried' or 'preserved' in the sediment. Empirical evidence and numerical modeling affirm that OC continues to be utilized by microbes even in very deep and ancient sediment[6–9]. Thus, the theoretical point at which OC degradation stops ($z_{ref}$) (under steady-state conditions, where $\delta OC/\delta z = 0$) does not exist. The continual nature of OC degradation becomes particularly apparent when OC degradation processes are framed over longer timescales.

II.  Specified reference depths (beneath the SWI) are highly variable between studies and can be from as little as 15 cm to tens of meters, sometimes pragmatically chosen to be the maximum depth of the sampled sediment core, and sometimes not reported[5,10,11].

III. Sediment depth can be an inadequate reference frame since biogeochemists and modelers are often concerned with understanding the fate of elements over particular timescales, rather than depth horizons.

IV.  There is limited comparability of $BE_{depth}$ between sites. A depth-based reference horizon ignores vastly different sedimentation rates between sites and thus the differing amounts of time that OC has been subject to degradation processes and other diagenetic alterations. For example, a sediment depth of 10 meters below the seafloor (mbsf) represents several thousand years of burial in

typical coastal sediments, and millions of years of burial in some abyssal regions (Fig. 2a). In addition, post-depositional reworking of sediments (e.g., due to bioturbation, erosion, tectonic events, and turbidity currents) may alter their position relative to sediments of other ages.

We argue that it is crucial to quantify OC burial by its depositional history and not simply by considering its depth beneath the seafloor. Studies should therefore consider using both explicitly-stated reference depth and age-horizons for quantifying carbon transfer through the ocean-sediment system.

BE can also be calculated on a temporal (rather than spatial) basis according to the flux of OC through a specified sediment age horizon ($BE_{age}$):

$$BE_{age} = \frac{100 \times F_{age}}{F_{SWI}} \tag{2}$$

Here, $F_{age}$ represents the OC flux through a specific sediment age horizon (defined by the transit time $t$ since deposition on the seafloor, e.g., $t = 100$ ka). $BE_{age}$ may be adjusted depending on the timescale of interest. The comparison of equivalent $BE_{age}$'s between different benthic settings may offer more consistency than using $BE_{depth}$'s—since the timescales of diagenetic alterations can be standardized using $BE_{age}$. However, we are aware of only one study that uses $BE_{age}$[12]. The limitation of this metric is that the age of a particular sediment horizon must be known or estimated (e.g., by using knowledge of past sedimentation rates, and chemical and biological age markers, whilst accounting for any post-depositional disturbances and sediment reworking).

We propose a new terminology, transfer efficiency ($T_{eff}$), for describing the fate of OC through clearly defined depth ($T_{eff,depth}$) or time ($T_{eff,age}$) horizons in marine sediments. The calculation of $T_{eff}$ is numerically equivalent to the calculation of BE, but it requires a precise definition of spatial or temporal reference horizons.

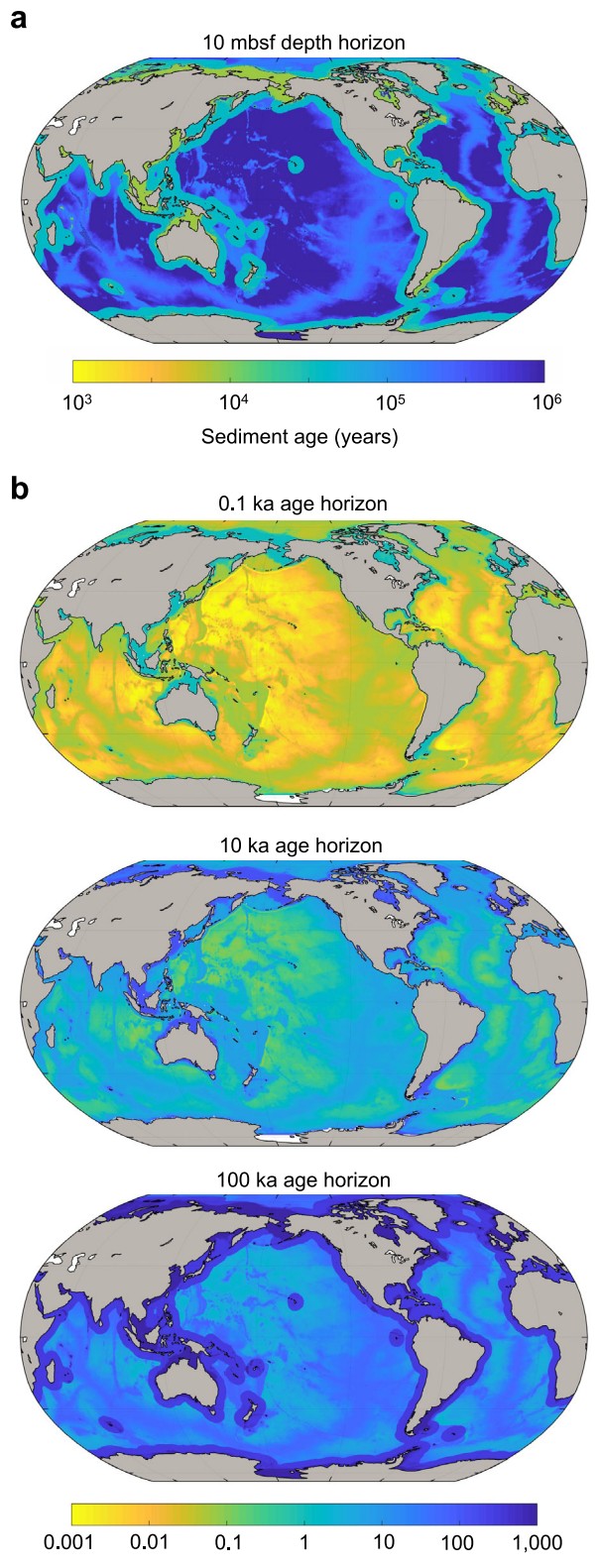

**Fig. 2 | Sediment ages and depths at specific horizons. a** Estimated sediment age (i.e., the time elapsed since its deposition at the SWI) at 10 mbsf. The estimated age of sediment at 10 mbsf varies by over three orders of magnitude globally. **b** Estimated sediment depth (mbsf) at horizons of equal sediment ages: 0.1 ka, 10 ka, and 100 ka. For a fixed sediment age, sediment depth beneath the seafloor varies globally by over three orders of magnitude.

$T_{eff,depth|age}$ is calculated according to:

$$T_{eff,depth|age}(SWI \rightarrow depth|age) = \frac{100 \times F_{depth|age}}{F_{SWI}} \qquad (3)$$

Where $depth|age$ represents the depth or age of the sediment horizon of interest, and $F_{depth|age}$ refers to the flux of organic carbon through that depth or age horizon. For example, $T_{eff,age}$ (SWI→100 ka) denotes the percentage of OC that has survived 100 ka of burial since its deposition at the SWI. The $T_{eff}$ terminology emphasizes that OC is not irreversibly buried but simply transits through a specified horizon. In addition, the precise specification of reference horizons enables comparability and upscaling between sites and studies.

We have carried out a series of calculations to illustrate how inconsistencies in BE metrics translate across different timescales, spatial scales, and depositional settings, using a spatially-resolved reaction transport model (RTM) for global sediments[13,14].

## Results and discussion

We estimate that the global OC burial rate at 0.11 mbsf (approximately equivalent to the bottom of the bioturbated zone) is between 0.114 and 0.202 Pg C yr⁻¹ (Fig. 3c, Supplementary Table 1). Our calculated OC burial rate is at the lower end of previous estimates (0.15–0.31 Pg C yr⁻¹)[15,16]. However, these previous estimates reported OC burial at unspecified depths beneath the seafloor. We estimate that the majority of OC is buried on the shelves (-0.105 Pg C yr⁻¹ at 0.11 mbsf). This is also the area with the highest uncertainty in estimated burial rates (between 0.079 and 0.135 Pg C yr⁻¹, Supplementary Table 1). Calculated $T_{eff}$'s are highest in abyssal sediments (Fig. 3a, b). However, the total OC burial flux in abyssal zones is low (between 0.024 and 0.048 Pg C yr⁻¹ at 0.11 mbsf, Fig. 3c, Supplementary Fig. 1, and Supplementary Table 1) since the OC concentrations in sediments in these regions (at the SWI and throughout the sediment depth profile) are generally much lower than in shelf and margin sediments[17,18]. The transit time ($t$) of sediment from deposition at the seafloor to 0.11 mbsf is also considerably longer in abyssal zones than in margin settings (Fig. 2).

Our results show that reference depths and ages (used to calculate $T_{eff,depth}$ and $T_{eff,age}$, respectively) greatly influence the total amount of carbon assumed to be buried in different depositional settings and across the entire seafloor (Fig. 3). Values of $T_{eff,depth}$ and $T_{eff,age}$, as well as the rates of OC burial, are most sensitive to reference depths and ages in shallower (<100 cm) and younger (<10 ka) sediments (Fig. 3b). These upper-most zones of sediments correspond to areas where OC degradation is fastest, due to the greater availability and preferential degradation of more reactive OC compounds (refs. 19, 20 and references therein). Therefore, precise specification and reporting of $T_{eff,depth}$ or $T_{eff,age}$ is particularly important for studies focusing on early diagenesis.

The clear specification of reference horizons used in the calculation of $T_{eff,depth}$'s or $T_{eff,age}$'s allows for adjustments to be made to these metrics based on the characteristic (temporal or spatial) scales of the problem considered. For example, to quantify the near-instantaneous interactions between the sediment and the ocean over annual timescales, the mixed-layer depth could be specified as a depth-horizon. Alternatively, a reference depth of meters to tens of meters below the seafloor could be specified to make estimates of OC budgets on millennial to million-year timescales. What determines a suitable reference depth or age depends on the specific application and problem to be addressed. However, studies reporting BE using a reference depth that is too shallow or a reference age that is too young may convey the impression that an unrealistically high amount of OC is buried (and presumed sequestered) in sediments. This is because OC continues to be degraded beyond these horizons (in deeper and older layers) (Fig. 3c).

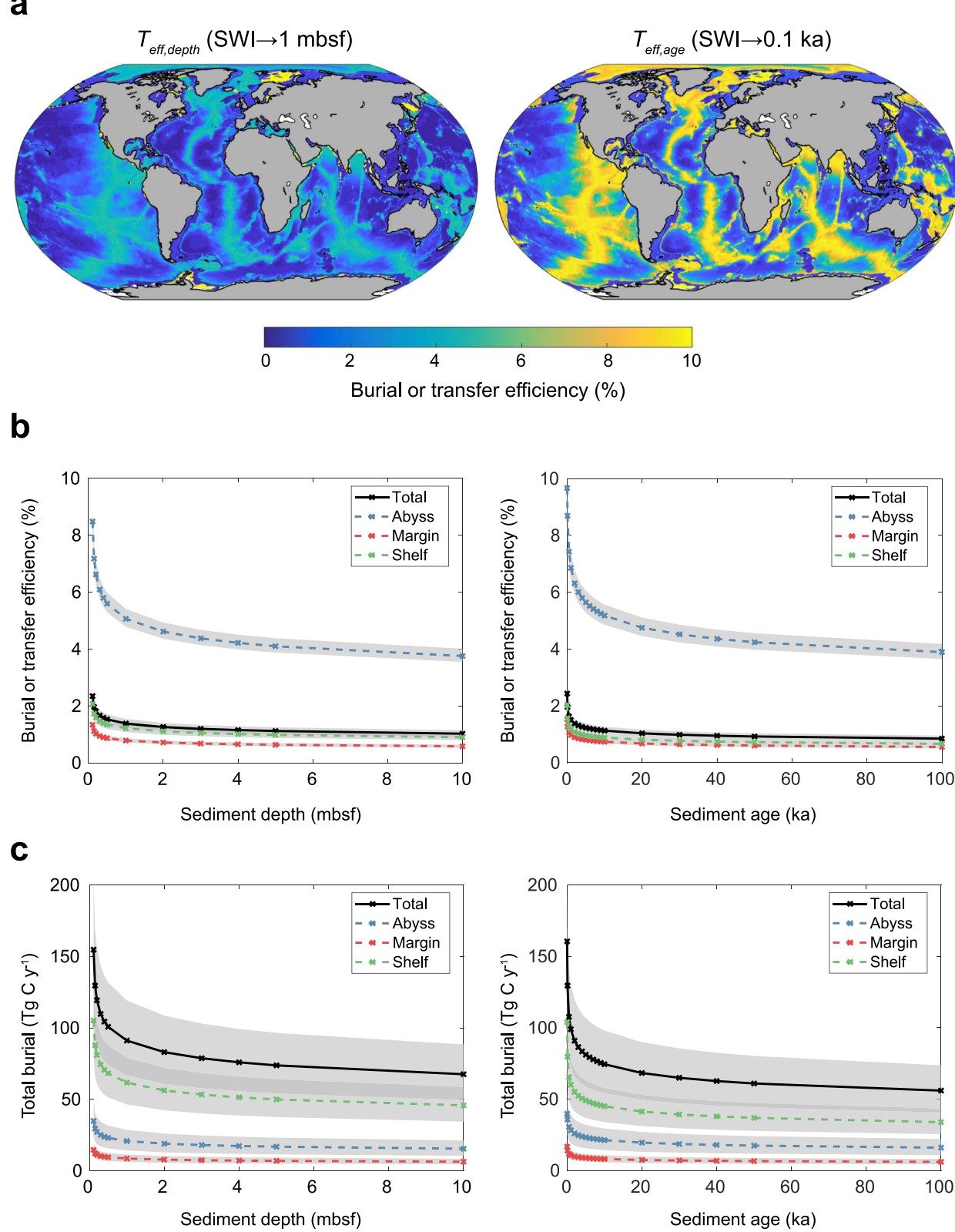

**Fig. 3 | Transfer and burial of organic carbon (OC) in marine sediments according to sediment depth and sediment age. a** Global maps of transfer efficiency from the sediment-water interface (SWI) to 1 mbsf ($T_{eff,depth}$ (SWI→1 mbsf), %) and from the SWI to 0.1 ka ($T_{eff,age}$ (SWI→0.1 ka), %). **b** Transfer (or burial) efficiency according to changes in the specified reference depth horizons and reference age horizons. **c** Total OC buried beyond specified sediment depths and ages. Gray shading in (**b**) and (**c**) represent uncertainty envelopes (±10% in $\varphi$ and $\omega$, see Supplementary Discussion).

Transfer efficiency ($T_{eff}$) (Eq. 3) is a more consistent and precise terminology for describing the fate of OC in marine sediments as it requires the specification of clearly defined depth ($T_{eff,depth}$) or age ($T_{eff,age}$) horizons within sediments, and is thus similar to how particulate OC fluxes are reported in the ocean[4,21,22]. This explicit description of OC burial according to a common pelagic-benthic framework (e.g., using $T_{eff}$ (SWI→100 ka) to describe the proportion of OC deposited that has survived 100,000 years of burial) ensures comparability of mass balance and flux calculations, as well as facilitating upscaling efforts between studies. In addition, the $T_{eff}$ notation emphasizes that OC is not irreversibly buried but simply transits through a given horizon and is thus removed from the OC pool at the particular spatial or temporal scale that is defined by $T_{eff}$. The $T_{eff}$ notation is thus more precise and consistent than BE in evaluating the transport and continual degradation of OC from the surface ocean to specific sediment depths and ages.

We propose that if $T_{eff}$'s are to be compared across settings, both depth and age should be considered. This is owing to (i) the enormous spatial heterogeneities in the age of sediment layers at fixed depths below the seafloor (Fig. 2a), and similarly variable sediment depths at fixed sediment age horizons (Fig. 2b), as well as (ii) the effect of changing reference depths and ages on $T_{eff}$ (Fig. 3). Studies should ideally consider both depth and time, i.e., when specifying a reference depth, time should be discussed (and vice-versa).

A complete mechanistic and quantitative understanding of the flux of OC through the sunlit ocean, its sinking and degradation in the water column, and its burial and degradation within sediments is necessary to understand global elemental cycling and its various roles on climate and the biosphere. The numerous biological, chemical, and physical processes controlling OC degradation and sequestration in sediments are highly heterogeneous over a wide range of spatial and temporal scales[19,20]. Moreover, the varying characteristics of diverse depositional settings (e.g., burial velocities, porosities, geochemistry) directly affect the timescales over which OC is degraded, and these must be considered when labeling OC as 'buried' or 'sequestered'. Reporting benthic OC fluxes according to a common spatially and temporally defined framework, $T_{eff}$, will ensure comparability between sites and studies, enable the integration between new measurements and existing data, and facilitate knowledge transfer and upscaling efforts. Ultimately, quantifying marine OC fluxes using consistent and robust metrics will enable an improved understanding of benthic-pelagic coupling and the role of marine carbon cycling in the Earth system.

## Methods

We use a one-dimensional RTM to calculate the burial and degradation rate of OC in sub-seafloor sediments[13,14], following the approach described in refs. 12, 23, and 24. The model is implemented on a $0.25° \times 0.25°$ resolution global grid. The geographical delineation of shelf, margin, and abyssal zones are adopted from ref. 12 (Supplementary Fig. 2). $T_{eff,depth}$ and $T_{eff,age}$ are calculated according to Eq. 3. The OC flux through a specific depth ($F_z$) is calculated according to:

$$F_z = -(1 - \varphi) \cdot \left( -D_b \cdot \frac{\partial OC(z)}{\partial z} + \omega \cdot OC(z) \right) \quad (4)$$

Where $OC$ is the concentration of organic carbon (g C cm$^{-3}$ dry sediment), $z$ is depth below the seafloor (cm), $\varphi$ represents sediment porosity and $\omega$ is the sedimentation rate (cm yr$^{-1}$).

### Organic carbon degradation dynamics

The one-dimensional conservation equation describing the transport and transformation of organic carbon ($OC$) in porous media is given by (e.g., refs. 25, 26):

$$\frac{\partial(1-\varphi)OC}{\partial t} = \frac{\partial}{\partial z}\left(D_b(1-\varphi)\frac{\partial OC}{\partial z}\right) - \frac{\partial(1-\varphi)\omega OC}{\partial z} + (1-\varphi)R_{OC} \quad (5)$$

Where $D_b$ (cm$^2$ yr$^{-1}$) denotes the bioturbation coefficient, and $R_{OC}$ (g C cm$^{-3}$ yr$^{-1}$) stands for the rate of organic carbon degradation.

We use a multi-G approximation of a reactive continuum model (RCM) to simulate organic carbon degradation kinetics (building on previous approaches[14,27]). The initial OC distribution of the RCM is constrained using the Gamma-distribution ($\Gamma$) and parameters $a$, $v$, and $k$:

$$f(k, 0) = \frac{a^v \cdot k^{v-1} \cdot \exp(-a \cdot k)}{\Gamma(v)} \quad (6)$$

Where $f(k, 0)$ determines the fraction of OC having a reactivity of $k$ at time zero. In Eq. 6, $a$ is the average lifetime (years) of the more reactive components of the OC mixture and $v$ is a dimensionless parameter determining the shape of the distribution near $k = 0$. The adjustable parameters $a$ and $v$ completely determine the shape of the initial distribution of OC compounds over the reactivity range and thus its overall reactivity. High $v$ and low $a$ values define an OC mixture dominated by compounds that are more rapidly degraded, and vice-versa. The Gamma distribution is defined (for any random variable, $x$) as:

$$\Gamma(v) = \int_0^\infty x^{v-1} \cdot \exp(-x)dx \quad (7)$$

The corresponding cumulative distribution function (CDF) which gives the fraction of total OC having a reactivity of $\leq k$ at time zero is defined as:

$$F(k, 0) = \frac{\Gamma(v, 0, a \cdot k)}{\Gamma(v)} = \frac{\int_0^{a \cdot k} x^{v-1} \cdot \exp(-x)dx}{\int_0^\infty x^{v-1} \cdot \exp(-x)dx} \quad (8)$$

Bulk OC, as constrained by the RCM above, is then approximated by 100 finite fractions each with their own first-order degradation rate constant, $k_i$. The reactivity range, here chosen to be $k = [10^{-15}, 10^{e_{max}}]$, with $e_{max} = -\log(a) + 2$ (ref. 12), is divided into $i = 100$ equal reactivity bins. The fraction of OC within the least reactive fraction $i = 1$ (i.e., with a degradation rate constant $k \leq 10^{-15}$ yr$^{-1}$) is calculated based on the lower incomplete Gamma function:

$$F_1\left(10^{-15}, 0\right) = \int_0^{a \cdot 10^{-15}} x^{v-1} \cdot \exp(-x)dx \quad (9)$$

The fraction, $i = 100$, of OC characterized by the highest reactivity is calculated based on the upper incomplete Gamma function:

$$F_{100}\left(10^{e_{max}}, 0\right) = \frac{\int_0^\infty x^{v-1} \cdot \exp(-x)dx - \int_0^{a \cdot e_{max}} x^{v-1} \cdot \exp(-x)dx}{\int_0^\infty x^{v-1} \cdot \exp(-x)dx} \quad (10)$$

The fractions of total OC within intermediate reactivity bins, $i \in [2, 99]$, are calculated with the CDF:

$$F_i(k_i, 0) = \frac{\Gamma(v, 0, a \cdot k_{i+1}) - \Gamma(v, 0, a \cdot k_i)}{\Gamma(v)}$$
$$= \frac{\int_0^{a \cdot k_{i+1}} x^{v-1} \cdot \exp(-x)dx - \int_0^{a \cdot k_i} x^{v-1} \cdot \exp(-x)dx}{\int_0^\infty x^{v-1} \cdot \exp(-x)dx} \quad (11)$$

All fractions $F_i$ add up to unity. The degradation rate of bulk OC can thus be calculated as:

$$R_{OC} = \sum_{i=1}^{100} k_i \cdot OC_i(z) \qquad (12)$$

Where $OC_i(0) = F_i \cdot OC_0$ assuming a known OC content at the SWI, $OC_0$. The derived degradation rate of OC, $R_{OC}$, was then used in Eq. 5 (i.e., the general conservation equation) to calculate OC concentrations, degradation and burial rates for the different sediment layers. For this purpose, the general conservation equation (Eq. 5) was solved analytically. Assuming steady-state conditions (i.e., $\frac{\partial OC}{\partial t} = 0$), and $D_b = 0$ for $z > z_{bio}$ (where $D_b$ represents the bioturbation coefficient (cm² year⁻¹), and $z_{bio}$ is the maximum depth of the bioturbated zone (cm)), the general solution of Eq. 5 for each organic carbon fraction $i$ in the bioturbated zone ($z \le z_{bio}$) is given by:

$$OC_i(z) = A_{1i} e^{(a_{1i}z)} + B_{1i} e^{(b_{1i}z)} \qquad (13)$$

And in the non-bioturbated zone ($z > z_{bio}$) by:

$$OC_i(z) = A_{2i} e^{(a_{2i}z)} \qquad (14)$$

With:

$$a_{1i} = \frac{\omega - \sqrt{(\omega^2 + 4D_b k_i)}}{2D_b} \qquad (15)$$

$$b_{1i} = \frac{\omega + \sqrt{(\omega^2 + 4D_b k_i)}}{2D_b} \qquad (16)$$

$$a_{2i} = \frac{-k_i}{\omega} \qquad (17)$$

The bulk OC concentration as a function of depth is then calculated as:

$$OC(z) = \sum_{i=1}^{100} OC_i(z) \qquad (18)$$

The integration constants $A_{1i}$, $B_{1i}$, and $A_{2i}$ are defined by chosen boundary conditions. Here, we apply a known OC concentration at the SWI and we assume continuity (in concentration and flux) across the bottom of the bioturbated zone, $z_{bio}$. The integration constants are thus calculated as:

$$B_{1i} = OC_0 - A_{1i} \qquad (19)$$

$$A_{2i} = \frac{A_{1i} \cdot \exp\left(a_{1i} \cdot \lim_{h \to 0}(z_{bio} - h)\right) + B_{1i} \cdot \exp\left(b_{1i} \cdot \lim_{h \to 0}(z_{bio} - h)\right)}{\exp\left(a_{2i} \cdot \lim_{h \to 0}(z_{bio} + h)\right)} \qquad (20)$$

$$A_{1i} = \frac{-B_{1i} b_{1i} \cdot \exp\left(b_{1i} \cdot \lim_{h \to 0}(z_{bio} - h)\right)}{a_{1i} \cdot \exp\left(a_{1i} \cdot \lim_{h \to 0}(z_{bio} - h)\right)} \qquad (21)$$

For $h > 0$ (see e.g., ref. 13 for details).

## Parameters and boundary conditions

For every grid cell we prescribe a particular concentration of organic carbon at the SWI, $OC_0$, and a set of parameter values (i.e., $\omega$, $D_b$, $\varphi$, and $a$) (Supplementary Fig. 3). Values of $OC_O$ are taken from ref. 18. Sedimentation rates, $\omega$, were calculated using an algorithm that correlates water depth and sedimentation rate according to a double logistic equation[28]. The bioturbation coefficient, $D_b$, also depends on water depth and follows the empirical relationship of ref. 29.

The porosities of sediments at the SWI were taken from ref. 30. We neglect sediment compaction and porosity changes (approximately 1/600 m⁻¹, ref. 31) in the upper 10 m of the sediment in order to find an analytical solution to Eq. 5. A comparison of the analytical solution with a numerical early diagenetic model with depth dependent porosity shows that porosity changes do not meaningfully affect our results[13].

A global parameter compilation[20] and inversely calculated RCM parameters[32,33] indicate that $v$ does not vary much between sites, while parameter $a$ can vary over orders of magnitude. Based on these results, we assume a constant $v$ value of $v = 0.125$ (characteristic of fresh organic matter). The values of parameter $a$ (i.e., shelf $a = 0.1$ yr, margin $a = 1.0$ yr, abyss $a = 20.0$ yr) were chosen to produce a realistic global OC burial rate that reflects the range observed in ref. 20. In order to account for lower OM reactivities and minimal bioturbation in low oxygen environments (e.g., refs. 29, 34, 35) we reduce the OM reactivity by an order of magnitude and set $z_{bio}$ equal to 1 cm in hypoxic seafloor zones (i.e., [O2] < 60 µM, according to bottom-water marine oxygen concentrations from the World Ocean Atlas 2018[36]).

## Model evaluation and sensitivity analysis

A detailed evaluation of the diagenetic model is provided in refs. 13, 14. We also compared our model output to five organic carbon (OC) profiles measured in sediment cores collected from different ocean depths and regions (Supplementary Table 2, Supplementary Discussion).

We performed a global sensitivity analysis to generate a ranking of the most important unknown model parameters (besides the reactivity of OM, i.e., $\varphi$, $\omega$, $z_{bio}$, and $D_b$) according to their relative contribution to the variability in model output (SI Fig. 4, Supplementary Discussion). The sensitivity analysis was used to generate uncertainty envelopes for our estimates of $T_{eff}$ and OC burial (Fig. 3) using a variability of ±10% of the two most influential parameters (i.e., $\omega$ and $\varphi$).

We used the method of ref. 37, also called the 'Elementary Effect Test' (EET[38]), which takes the mean of $r$ finite differences (also called the 'Elementary Effects' or EEs) as a measure of global sensitivity of input parameter $i$:

$$S_i = \frac{1}{r} \sum_{j=1}^{r} EE^j = \frac{1}{r} \frac{\sum_{j=1}^{r} g\left(x_1^j, ..., x_i^j + \Delta_i^j, ... x_M^j\right) - g\left(x_1^j, ..., x_i^j, ... x_M^j\right)}{\Delta_i^j} \qquad (22)$$

Where $g()$ is our diagenetic model, OMEN-SED, that maps the vector of the input factors $x^j = (x_1^j, ..., x_M^j)$ into the output space—here the simulated OC burial rates at 1 mbsf. $\Delta_i^j$ represents the variation of the input parameter $i$. We compute the standard deviation of the EEs, which measures the degree of interaction of input parameter $i$ with the other input parameters. Both sensitivity indices are relative measures, hence their values do not have a specific meaning and can only be used to rank the influence of the input parameters. As a strategy to select the parameter vectors $x^j$ ($j = 1, ..., r$) and the input variations $\Delta_i$ for the investigated model parameters ($M = 4$), we used the Latin hypercube sampling approach as implemented in the Sensitivity Analysis for Everyone (SAFE) MATLAB toolbox[39]. For $z_{bio}$ we explored a range between 1 and 15 cm. For $\varphi$, $\omega$, and $D_b$ we varied the nominal values in each grid cell by up to 20%.

The calculations of the mean and standard deviation of the EEs of $M$ input parameters requires $N = r \cdot (M + 1)$ model evaluations. To assess

the robustness of our sensitivity indices, i.e., to analyze if they are independent of the specific input–output sample, we calculated bootstrapping-based confidence limits of the indices. Following recommendations in the literature (e.g., ref. 40), we calculated $r = 30$ finite differences, which is sufficient to differentiate between influential and non-influential parameters, to calculate reasonable confidence bounds of the sensitivity indices. In total we ran $N = r \cdot (M + 1) = 150$ global model simulations with different input parameter values.

## Data availability
Data from this study is available at https://zenodo.org/badge/latestdoi/566835035.

## Code availability
The version of the model code used in this study is tagged as release v1.0 and is available at https://zenodo.org/badge/latestdoi/566835035. Necessary boundary condition files and observational data are included as part of the code release.

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

## Acknowledgements
We acknowledge funding from NERC (NE/T010967/1) (J.A.B.), the Alexander von Humboldt Foundation (J.A.B.), the Human Frontier Science Program (J.A.B.), the Simons Foundation (653829) (D.H.), C-DEBI (NSF OCE0939564) (D.E.L.), NASA-NSF Origins of Life Ideas Lab (NNN13D466T) (D.E.L.), NASA Habitable Worlds (80NSSC20K0228) (D.E.L.), and BELSPO FedtWin program RECAP (S.A.).

## Author contributions
J.A.B. and D.H. contributed equally to this work. J.A.B. conceived the study. J.A.B., D.H., and S.A. designed the research. D.H. conducted the simulations. J.A.B., D.H., D.E.L., and S.A. analyzed model output. J.A.B. and D.H. wrote the manuscript with input from D.E.L. and S.A.

## Competing interests
The authors declare no competing interests.
