## [Peer Review File · Nature Communications]

Transfer efficiency of organic carbon in marine sedimentsREVIEWER COMMENTS

Reviewer #1 (Remarks to the Author):

The biological carbon pump describes the source and fate of organic carbon in the oceans. Accelerating climate change and the likelihood of application of new carbon capture technologies make it critical that scientists more accurately quantify predictions of buried organic carbon and the rate at which that carbon is remineralized at different depths. Bradley et al. introduce an updated metric to replace the metric of burial efficiency (BE). BE is the estimate of the percentage of organic carbon that is deposited on the seafloor that becomes buried beyond some specified reference depth. Since this depth is often arbitrarily defined and inconsistent, the notion of BE is misleading and makes comparison between studies problematic. Bradley et al. suggest in this paper that "transfer efficiency" is a better metric because it takes into account the possible orders of magnitude disparities in burial of organic carbon between equivalent sediment depths and ages in different locations. This results in vastly different estimations of burial efficiencies (BE) depending on what reference horizon is used. While this paper may at first glance seem to be a nuanced improvement over the BE estimations made in previous publications by these and other authors, e.g., La Rowe et al. 2020, that is used in global diagenetic models to predict carbon burial, the authors point out one of the most serious flaws in the BE predictions, which is a lack of a reference age or depth horizon. This is a serious flaw, which leads to BE predictions even for closely located samples that are inconsistent, and misleading. At a time when sediment burial is often equated with "permanently sequestered" in conversations about carbon capture strategies to mitigate global warming, these kinds of misconceptions can lead to terrible policy decisions. The readers of this journal are keenly interested to be on top of this topic at a time when decision makers worldwide are having conversations about carbon "storage" that are loaded with assumptions about "permanently stored", or "sequestered" carbon. The calculation these authors propose, transfer efficiency, is numerically equivalent to BE, but requires the specification of precise reference ages or depths, and takes into account the reality that buried carbon continues to be remineralized in the subsurface biosphere, albeit at much slower (but not insignificant) rates. I have some comments for the authors to consider.

1. I think it is important to make it clearer to readers that in spite of the excellent previous works by these authors in this area, that current models used to predict different global pools of carbon as well as rates of its remineralization in sediments in different oceanic regimes involve a LOT of unknowns, or are at least based on limited data. The first location in the paper that I feel this could be clarified is the Figure 1 legend. The authors go on to point out almost all my concerns in the text after Figure 1, but figures in high profile papers can be taken and used out of context easily. There is a paucity of data for benthic fluxes of OC through depths, and so the estimates in the figure are simply those, estimates of global deposition and burial. The dashed line representing concentration of OC also implies that it goes to zero at some depth. While this may be true in some locations, and the depth is not specified intentionally, this may not happen in many locations and depths of relevance to the discussions of carbon burial and global climate. The literature is populating slowly but surely with studies that show biological activities in the sedimentary as well as lithified deep biosphere, and the degree to which microbiota are active appears to vary significantly based on local conditions (including deep biosphere fluid circulation, temperature, etc.). I think it would be prudent to point out that this figure reflects a current paradigm based on limited data. The figure legend title could perhaps say "Conceptual framework for deposition and burial of OC in marine sediments." And there could be some notation about OC and how it may never get to zero.
2. Line 75-79: please consider adding a statement about other site-related differences that can drastically affect comparability of sediment horizons from different sites when using depth alone to define a horizon. For instance, continental shelf (or other location) slope sloughs of sediment, bottom currents, storm events, bioturbation, tectonic activities, all of which can make it impossible to compare depth horizons between sites because they may be significantly different ages if top layers are removed or sediment loads are deposited during sloughs, etc.
3. Line 100-103: this kind of estimation of age based on sedimentation rate is only viable in the absence of disturbances. Please note.
4. Line 112-113: Instead of saying "our results reveal" please consider rephrasing to make it clearer the model you are using to make this prediction of integrated OC burial, that it is an

ESTIMATE, and also, please consider adding a line noting the assumptions in that model that may change in the future when we have more data on benthic remineralization rates at different depths/sediment ages in different hydrodynamic settings. On lines 159 onward, the authors imply this by stating that we need a "complete mechanistic and quantitative understanding of carbon fixation in the sunlit ocean, its sinking and degradation through the water column and its burial and degradation in sediments is necessary to understand global elemental cycling and its roles on climate." Rephrasing statements as on line 112-113 to avoid readers quoting that statement and presenting it as all is known here, is wise. The last paragraph starting on Line 159-170 is a great place to loop back to your estimation of OC burial to make it clear that firmer estimates will come with more data.

Reviewer #2 (Remarks to the Author):

This manuscript presents a discussion upon the nuances of quantifying carbon burial efficiency in marine sediments and the importance in defining what is meant by "carbon burial". The authors attempt to develop a framework can be used to operationally define carbon burial so that future work on this topic can be better viewed in the context of global biogeochemical cycles. The authors then use a global diagenetic model to demonstrate the likely ranges of carbon burial efficiency.

I think authors bring up important points and this is an important discussion is necessary for understanding the long-term carbon cycles and as such would be important to a wide range of researchers (biogeosciences, earth systems, climate scientist) beyond simply the narrow field of early diagenesis. For example, even the consideration of cohesive sediments as a form of so called "Blue Carbon" or seafloor trawling as a source of potential source of CO₂ due to enhanced sediment remineralization decreased carbon burial. As researchers from a wider variety of backgrounds and research fields begin to consider implications of sediment carbon burial and efficiency. It is important that these terms are understood and defined with a proper understanding of early diagenesis or misunderstanding incorrect interpretations will arise. I think for this reason this paper is important discussion that is suitable for Nature Communications, however I think there are a few revisions which I would urge the authors to consider.

I like the framework the authors present for defining carbon burial both in terms of either the depth or timescale of burial. I think this is a major improvement over the general way it is currently talked about. However, I think they can take this discussion further. There is a relationship between both the timescale and length scale of burial that I think the authors should explore more (they do touch on it). The burial efficiency depends not just on remineralized and burial of particulate carbon but also by the return of DIC to the overlying water which would be set by the timescale of DIC diffusion from deep in the sediments. I wonder if this provides a way to link both the burial efficiency timescale and depths. I think this is the kind of question a diagenetic model such as theirs is ideally set up to consider.

A bit more discussion explicit discussion about what the appropriate timescales or depths of carbon burial are relevant for the global climate cycle. The authors use 100ka as an example timescale, which is logical when considering feedback on climate, however other issues might want to consider other timescales of burial (i.e. the sediment role in coastal eutrophication). I think this would help fill out the discussion some more and make the paper appealing to a wider audience.

With regards to the diagenetic model, I would like to see more detail on the how they validated their model. For example, I would like to see plots showing how their modeled sediment organic carbon profiles compare to measured profiles in different marine environments (i.e. abyssal plains, shelves, upwelling zones etc.).

I would like to see a model sensitively study indicating how sensitive the model is to the functions parameterizing sedimentation rate and bioturbation, since the scatter on these relationships, particularly bioturbation, can vary by an order of magnitude.

Reviewer #3 (Remarks to the Author):

Bradley, Hülse et al., identify that the commonly used "Burial Efficiency" metric (the percentage of organic carbon hitting the seafloor that persists beyond a given depth horizon), has a number of limitations in its application and definition. In particular they use a global diagenetic model to demonstrate that depth horizons can be problematic to define burial as different sediment settings have significantly different sedimentation rates and concentrations of organic carbon. The authors suggest an updated terminology of "Transfer Efficiency" that can reflect a depth or age horizon and which better reflects the interpretation of burial.

This is a concise and very well written manuscript which is to the point! The authors have identified and demonstrated some clear limitations with the use of Burial Efficiency as a metric . However, the way in which the Transfer Efficiency metric is defined and presented is not as clear as it should be. Overall, I think the manuscript is valuable and accessible to a large community.

General Comments

Definition of Transfer Efficiency - The definition of transfer efficiency towards the end of the manuscript (lines 139 - 154) led to some confusion when reading the manuscript as the term is used prior to this in Figure 3. Additionally, it took a few reads to really understand that the definition incorporated the two previous definitions of Burial Efficiency rather than a totally new metric. In part this was the language used and the fact that the definition appeared much later than the discussion of Burial Efficiency. I would suggest: 1) defining Transfer Efficiency earlier on in the text around equation 2 and before the Results and Discussion. This helps clarify the key arguments of the manuscript otherwise the transfer efficiency metric appears more like a renaming. Additionally, this directly follows the second Burial Efficiency definition and builds on that otherwise I had forgotten it was used by one paper only! ...and/or... 2) clarifying the existing definition text (see specific comments below) and updating the use of the term in Figure 3 so that the reader is not searching ahead, e.g., use two specific Burial efficiency definitions and potentially add your equivalent fully expanded transfer efficiency metric alongside.

In general I think the new definition of transfer efficiency is useful for the reasons outlined by the authors. However, I thought it's worth noting a perspective on the use of the term from the marine biological pump community. "Transfer Efficiency" is, as far as I'm aware, solely used to describe the POC flux reaching a depth horizon whereas a separate term, "Sequestration Depth", is used to describe the flux reaching an age horizon (age here is defined as the average residence time of a dissolved tracer at this depth horizon) (Weber et al., 2018; Boyd et al., 2019). I don't think this is an issue for the manuscript but could be clarified.

Specific Comments

Line 32: "contribute" should be "contributed"

Line 52: to be asymptotic $\partial OC/\partial z = 0$ should be approximately zero or practically defined as as smaller than some value?

Lines 72 - 74: This needs a clarifying sentence along the lines of "sediment depth represents a specific timescale of burial that is a function of local sedimentation rates".

Line 79: A reference to Figure 2a seems most appropriate.

Figure 2: Panel b is not described in text. The panels are largely an inverse of panel a, which makes a lot of sense, but without any description I'm not sure what I'm meant to understand from the three panels.

Lines 100 - 103: is this a significant limitation or is this readily calculated? How reliable are estimates using the sedimentation rate?

Lines 117 - 118: "most sensitive to reference depths and ages..." could you briefly outline why

transfer efficiency is changing most rapidly here?

Lines 120 - 121: is it possible to provide more specific, or at least indicative, definitions of "shallow", "young", and "unrealistically high".

Figure 3: it would help to see the abyss, margin and shelf areas delineated on a map if that's appropriate/possible?

Figure 3: It would be great to use the transfer efficiency term on the figure as you have defined it, e.g., referring to specific depth or age horizons, given that this is a key point of the manuscript!

Lines 139 - 141: "more robust metric" led me to think this was something numerically unique from the two burial efficiency definitions. I would suggest the transfer efficiency is a new terminology that is more flexible and precise in its definition than burial efficiency.

Eqn 3: I have to admit I did not know the symbol for OR used in the equation! Could there be a quick definition in the text?

Response to reviewers' comments on 'Closing the Biological Carbon Pump: Transfer Efficiency of Organic Carbon in Marine Sediments.'

We would like to thank the reviewers for their helpful and constructive comments and suggestions, which helped improve the manuscript significantly. We have addressed all concerns that were raised.

Please find below our responses to the specific points. Line numbers refer to lines when track changes are hidden.

REVIEWER COMMENTS

Reviewer #1 (Remarks to the Author):

The biological carbon pump describes the source and fate of organic carbon in the oceans. Accelerating climate change and the likelihood of application of new carbon capture technologies make it critical that scientists more accurately quantify predictions of buried organic carbon and the rate at which that carbon is remineralized at different depths. Bradley et al. introduce an updated metric to replace the metric of burial efficiency (BE). BE is the estimate of the percentage of organic carbon that is deposited on the seafloor that becomes buried beyond some specified reference depth. Since this depth is often arbitrarily defined and inconsistent, the notion of BE is misleading and makes comparison between studies problematic. Bradley et al. suggest in this paper that “transfer efficiency” is a better metric because it takes into account the possible orders of magnitude disparities in burial of organic carbon between equivalent sediment depths and ages in different locations. This results in vastly different estimations of burial efficiencies (BE) depending on what reference horizon is used. While this paper may at first glance seem to be a nuanced improvement over the BE estimations made in previous publications by these and other authors, e.g., La Rowe et al. 2020, that is used in global diagenetic models to predict carbon burial, the authors point out one of the most serious flaws in the BE predictions, which is a lack of a reference age or depth horizon. This is a serious flaw, which leads to BE predictions even for closely located samples that are inconsistent, and misleading. At a time when sediment burial is often equated with “permanently sequestered” in conversations about carbon capture strategies to mitigate global warming, these kinds of misconceptions can lead to terrible policy decisions. The readers of this journal are keenly interested to be on top of this topic at a time when decision makers worldwide are having conversations about carbon “storage” that are loaded with assumptions about “permanently stored”, or “sequestered” carbon. The calculation these authors propose, transfer efficiency, is numerically equivalent to BE, but requires the specification of precise reference ages or depths, and takes into account the reality that buried carbon continues to be remineralized in the subsurface biosphere, albeit at much slower (but not insignificant) rates. I have some comments for the authors to consider.

I think it is important to make it clearer to readers that in spite of the excellent previous works by these authors in this area, that current models used to predict different global pools of carbon as well as rates of its remineralization in sediments in different oceanic regimes involve a LOT of unknowns, or are at least based on limited data. The first location in the paper that I feel this could be clarified is the Figure 1 legend. The authors go on to point out almost all my concerns in the text after Figure 1, but figures in high profile papers can be taken and used out of context easily. There is a paucity of data for benthic fluxes of OC through depths, and so the estimates in the figure are simply those, estimates of global deposition and burial.

Whilst the central focus of this manuscript is to encourage a revision in the metrics used to quantify OC burial (using global diagenetic modelling to illustrate this point), we agree that it is nevertheless important to ensure that the reporting of our model simulation results makes clear the limitations, potential errors and uncertainty. We have modified the phrasing of the Figure 1 caption according to the reviewer's suggestions, and also the phrasing used to describe and discuss model results - making clear that our calculations are associated with various sources of uncertainty and unknowns. We have also evaluated the sensitivity of our model estimates to key parameters (SI Figures 4 and 6), and added uncertainty envelopes in Figure 3 (details on sensitivity and uncertainty analyses are provided in the SI and the responses to specific reviewer comments below).

The dashed line representing concentration of OC also implies that it goes to zero at some depth. While this may be true in some locations, and the depth is not specified intentionally, this may not happen in many locations and depths of relevance to the discussions of carbon burial and global climate. The literature is populating slowly but surely with studies that show biological activities in the sedimentary as well as lithified deep biosphere, and the degree to which microbiota are active appears to vary significantly based on local conditions (including deep biosphere fluid circulation, temperature, etc.). I think it would be prudent to point out that this figure reflects a current paradigm based on limited data. The figure legend title could perhaps say "Conceptual framework for deposition and burial of OC in marine sediments." And there could be some notation about OC and how it may never get to zero.

We have made modifications to Figure 1 to address this point:

- Dashed line does not go to zero
- Expanded the dashed lines with a shaded region to provide an indicative range of OC concentrations.
- Modified the figure legend to incorporate suggestions made by the reviewer.

Figure 1. Schematic of the deposition and burial of organic carbon (OC) in idealized marine sediments in shelf and abyssal zones. The dashed black lines represent illustrative OC concentrations ($[OC]$) for shelf and abyssal sediments at certain depths and their equivalent (exemplar) ages, and the dark shading represents possible variability in OC concentration between sites. The red arrows indicate the flux

of OC through the sediment water interface (F_{SWI}), as well as through specific depth or age layers (F_{depth} (e.g. $F_{0.5 m}$), and F_{age} (e.g. $F_{0.1 ka}$), respectively). The widths of the red arrows represent the magnitude of the OC fluxes through those layers. In shelf sediments, OC is rapidly degraded near the sediment-water interface, where shallow sediment depths correspond to short burial times. Conversely, in abyssal sediments, low concentrations of OC persist over long timescales. In the deep ocean, sediments buried at shallow depths (beneath the SWI) have much longer burial times than sediments of an equivalent depth (beneath the SWI) in shallow water. This is due to low sedimentation rates in abyssal zones.

Furthermore, we have included uncertainty envelopes to our estimates of OC burial fluxes and burial efficiencies (see Figure 3). These represent the main uncertainties in our model parameters and are based on a global sensitivity analysis which we included into the SI.

Figure 3. Transfer and burial of organic carbon (OC) in marine sediments according to sediment depth and sediment age. (a) Global maps of transfer efficiency at 1 mbsf ($T_{eff,depth}$ (SWI→1 mbsf, %)) and at 0.1 ka ($T_{eff,age}$ (SWI→0.1 ka, %)), (b) Transfer (or burial) efficiency according to changes in the specified reference depth horizons and reference age horizons, (c) Total OC buried beyond specified sediment depths and ages. Grey shading in (b) and (c) represent uncertainty envelopes ($\pm 10\%$ in φ and ω , see Supplementary Information).

Line 75-79: please consider adding a statement about other site-related differences that can drastically affect comparability of sediment horizons from different sites when using depth alone to define a horizon. For instance, continental shelf (or other location) slope sloughs of sediment, bottom currents, storm events, bioturbation, tectonic activities, all of which can make it impossible to compare depth horizons between sites because they may be significantly different ages if top layers are removed or sediment loads are deposited during sloughs, etc.

We have noted this by adding the following text:

In addition, post-depositional reworking of sediments (e.g., due to bioturbation, erosion, tectonic events, and turbidity currents) may alter their position relative to sediments of other ages.

(Lines 98-100)

Line 100-103: this kind of estimation of age based on sedimentation rate is only viable in the absence of disturbances. Please note.

We have rephrased this sentence to address this point:

The limitation of this metric is that the age of a particular sediment horizon must be known or estimated (e.g., by using knowledge of past sedimentation rates, and chemical and biological age markers, whilst accounting for any post-depositional disturbances and sediment re-working).

(Lines 123-126)

Line 112-113: Instead of saying “our results reveal” please consider rephrasing to make it clearer the model you are using to make this prediction of integrated OC burial, that it is an ESTIMATE, and also, please consider adding a line noting the assumptions in that model that may change in the future when we have more data on benthic remineralization rates at different depths/sediment ages in different hydrodynamic settings. On lines 159 onward, the authors imply this by stating that we need a “complete mechanistic and quantitative understanding of carbon fixation in the sunlit ocean, its sinking and degradation through the water column and its burial and degradation in sediments is necessary to understand global elemental cycling and its roles on climate.” Rephrasing statements as on line 112-113 to avoid readers quoting that statement and presenting it as all is known here, is wise. The last paragraph starting on Line 159-170 is a great place to loop back to your estimation of OC burial to make it clear that firmer estimates will come with more data.

We have modified the phrasing of statements in question (lines 112-113 in the original manuscript), as well as elsewhere in the manuscript to clarify that our model simulations of OC degradation and burial rates are estimates.

We estimate that the global OC burial rate at 0.11 mbsf (approximately equivalent to the bottom of the bioturbated zone) is between 0.114 and 0.202 Pg C yr⁻¹ (Figure 3c, SI Table 1). Our calculated OC burial rate is at the lower end of previous estimates (0.15 – 0.31 Pg C yr⁻¹)^{15,16}. However, these previous estimates reported OC burial at unspecified depths beneath the seafloor.

(Lines 146-149)

We have also added a global sensitivity analysis (Supplementary Information), and uncertainty envelopes in Figure 3. This further underscores that our calculations are estimates based on the best current knowledge.

We also, as suggested, stress in the final paragraph how the integration of new data with existing data will bring about an improved understanding of the benthic carbon cycle.

Reporting benthic OC fluxes according to a common spatially and temporally defined framework, T_{eff} , will ensure comparability between sites and studies, enable the integration between new measurements and existing data, and facilitate knowledge transfer and upscaling efforts. Ultimately, quantifying marine OC fluxes using consistent and robust metrics will enable an improved understanding of benthic-pelagic coupling and the role of marine carbon cycling in the Earth system.

(Lines 215-220)

Reviewer #2 (Remarks to the Author): This manuscript presents a discussion upon the nuances of quantifying carbon burial efficiency in marine sediments and the importance in defining what is meant by “carbon burial”. The authors attempt to develop a framework can be used to operationally define carbon burial so that future work on this topic can be better viewed in the context of global biogeochemical cycles. The authors then use a global diagenetic model to demonstrate the likely ranges of carbon burial efficiency.

I think authors bring up important points and this is an important discussion is necessary for understanding the long-term carbon cycles and as such would be important to a wide range of researchers (biogeosciences, earth systems, climate scientist) beyond simply the narrow field of early diagenesis. For example, even the consideration of cohesive sediments as a form of so called “Blue Carbon” or seafloor trawling as a source of potential source of CO₂ due to enhanced sediment remineralization decreased carbon burial. As researchers from a wider variety of backgrounds and research fields begin to consider implications of sediment carbon burial and efficiency. It is important that these terms are understood and defined with a proper understanding of early diagenesis or misunderstanding incorrect interpretations will arise. I think for this reason this paper is important discussion that is suitable for Nature Communications, however I think there are a few revisions which I would urge the authors to consider.

I like the framework the authors present for defining carbon burial both in terms of either the depth or timescale of burial. I think this is a major improvement over the general way it is currently talked about. However, I think they can take this discussion further.

We have addressed this suggestion in the revised manuscript by slightly expanding the discussion. Nevertheless, we have not added substantial additional text to the discussion. It was our original intention to write a short and focused manuscript (as noted in the Cover Letter to the Editor: ‘The article is deliberately short to articulate our main point clearly and concisely.’) and the comments we have received from reviewers – in particular, Reviewer #3 – suggest that this is effective (*‘This is a concise and very well written manuscript which is to the point!’*).

There is a relationship between both the timescale and length scale of burial that I think the authors should explore more (they do touch on it). The burial efficiency depends not just on remineralized and burial of particulate carbon but also by the return of DIC to the overlying water which would be set by the timescale of DIC diffusion from deep in the sediments. I wonder if this provides a way to link both the burial efficiency timescale and depths. I think this is the kind of question a diagenetic model such as theirs is ideally set up to consider.

The timescale/length-scale relationship is one of the critical points of our manuscript (alongside the specification of precise reference horizons). We have amended the revised manuscript to more directly link timescales and length scales in places, such as:

The clear specification of reference horizons used in the calculation of $T_{eff,depth}$'s or $T_{eff,age}$'s allows for adjustments to be made to these metrics based on the characteristic (temporal or spatial) scales of the problem considered. For example, to quantify the near-instantaneous interactions between the sediment and the ocean over annual timescales, the mixed-layer depth could be specified as a depth-horizon. Alternatively, a reference depth of meters to tens of meters below the seafloor could be specified to make estimates of OC budgets on millennial to million-year timescales. What determines a suitable reference depth or age depends on the specific application and problem to be addressed. However, studies reporting BE using a reference depth that is too shallow or a reference age that is too young may convey the impression that an unrealistically high amount of OC is buried (and presumed sequestered) in sediments. This is because OC continues to be degraded beyond these horizons (in deeper and older layers) (Figure 3c).

(Lines 168-178)

Indeed, the timescale of sequestration of carbon in sediments depends not only on OC burial efficiency, but also on the timescale by which the DIC that is produced in the sediments diffuses back to the overlying bottom waters. The metrics with which to quantify and report the return flux of DIC are important but are an entirely different problem which goes well beyond the present manuscript. Moreover, our diagenetic model assumes steady-state is not appropriate to answer this question. Nevertheless, we have carried out additional simplified calculations to make first order estimates of the DIC return timescales under specific conditions (see Figure R1, below).

Here, we use a molecular diffusion coefficient for CO₂ at 0°C of $D^0 = 1.92E-5 \text{ cm}^2 \text{ sec}^{-1}$ (Boudreau, 1997) and account for sediment porosity (ϕ) and tortuosity (F) effects (but neglect irrigation in the upper bioturbated layer) to calculate the specific molecular diffusion coefficient for CO₂ in the sediment:

$$D_{mol}^{CO_2} = \frac{D^0}{\phi \cdot F}$$

Tortuosity can be expressed in terms of porosity as (Ullman and Aller, 1982):

$$F = \frac{1}{\phi^m}$$

With the exponent m varying according to the type of sediment (here $m = 3$ is used representing muddy sediments with high porosity). We then calculate the DIC return timescales from sub-seafloor depths of up to (a) 1 m, (b) 10 m, and (c) 30 m, and a range of sediment porosities. Our simple estimates show that, in theory, the timescales for DIC return

to the overlying bottom waters can differ by more than 3 orders of magnitude, from years to decades for DIC produced at 1 mbsf, to tens of thousands of years for DIC produced at 30 mbsf. However, our simplified calculation of DIC return timescales does not include critical processes such as carbonate precipitation, or a representation of the pH system. Capturing these processes would be necessary for a global scale assessment of DIC return fluxes – a problem that should be dealt with in another manuscript as we intended to focus our short article on metrics for OC burial.

Figure R1: DIC return timescale (years) considering the diffusion of DIC (through the sediment column) produced at various subsurface depths (a-c) and porosities (blue, red, green).

A bit more discussion explicit discussion about what the appropriate timescales or depths of carbon burial are relevant for the global climate cycle. The authors use 100ka as an example timescale, which is logical when considering feedback on climate, however other issues might want to consider other timescales of burial (i.e. the sediment role in coastal eutrophication). I think this would help fill out the discussion some more and make the paper appealing to a wider audience.

We have modified the manuscript to specify some equivalent spatial and temporal scales/use cases, and we also show the equivalent depth-age horizons (10 mbsf, 0.1 ka, 10 ka, 100 ka) in Figure 2.

The clear specification of reference horizons used in the calculation of $T_{eff,depth}$'s or $T_{eff,age}$'s allows for adjustments to be made to these metrics based on the characteristic (temporal or spatial) scales of the problem considered. For example, to quantify the near-instantaneous interactions between the sediment and the ocean over annual timescales, the mixed-layer depth could be specified as a depth-horizon. Alternatively,

a reference depth of meters to tens of meters below the seafloor could be specified to make estimates of OC budgets on millennial to million-year timescales. What determines a suitable reference depth or age depends on the specific application and problem to be addressed. However, studies reporting BE using a reference depth that is too shallow or a reference age that is too young may convey the impression that an unrealistically high amount of OC is buried (and presumed sequestered) in sediments. This is because OC continues to be degraded beyond these horizons (in deeper and older layers) (Figure 3c).

(Lines 168-178)

With regards to the diagenetic model, I would like to see more detail on the how they validated their model. For example, I would like to see plots showing how their modeled sediment organic carbon profiles compare to measured profiles in different marine environments (i.e. abyssal plains, shelves, upwelling zones etc.).

We have added a new figure in the SI (SI Figure 5) showing how well our model reproduces measured TOC profiles from sediment cores collected in different depositional settings/water depths (i.e., shelf, margin, and abyss sediments) and regions (i.e., Iberian shelf and margin, Santa Barbara margin, North Atlantic and South Pacific Gyres). Importantly, these simulations were carried out without tuning model parameters to local conditions.

We note here (and also in the SI) that these profiles serve to show that our model is able to simulate the general trends in the marine sediments. However, we do not expect that our model simulates all local sediment POC measurements for all sites, largely because (i) profiles from individual cores are subject to processes that are not relevant across larger spatial and temporal scales, (ii) core profile data do not reflect the spatial heterogeneity of a site or a region, and so the appropriateness of a particular measurement in its reflection of a wider area or process is not known, and (iii) individual data suffer measurement accuracy and bias.

In general, continuum models of organic matter degradation, such as the RCM used here, are theoretically derived and rest on a large body of organic matter depth profiles and rate measurements from a wide range of marine settings that show a decrease of organic matter reactivity with sediment depth and/or burial age. We have included more information on the general RCM approach and the specific validation of our RCM approximation to the Supplementary information.

Indeed, it is important to ensure that our reporting of our model simulation results includes an account of the unknowns, potential error and uncertainty, and thus we have modified our phrasing (clarifying that our results represent estimates that are associated with various sources of uncertainty and unknowns) and we have also quantified, visualized and discussed uncertainty in the revised manuscript (see below).

SI Figure 5: Model-data comparison. Modelled (blue lines) and measured (shaded markers) OC concentration profiles for sediment cores. from: (a) the Iberian Shelf¹⁸, (b) the Santa Barbara Margin¹⁹, (c) the Iberian Margin¹⁸, (d) the North Atlantic Abyss⁶, and (e) the South Pacific Abyss⁶. zone and water depth). The panel headings indicated depositional environment and water depth. Further details on core measurements are provided in SI Table 2. The green dashed lines indicate the depth of the bioturbation zone.

I would like to see a model sensitively study indicating how sensitive the model is to the functions parameterizing sedimentation rate and bioturbation, since the scatter on these relationships, particularly bioturbation, can vary by an order of magnitude.

We have added a global sensitivity analysis to the SI in order to rank the importance of four unknown model parameters (z_{bio} , D_b , ϕ , ω) according to their relative contribution to the output variability. We ran 150 global simulations with different parameter combinations and from the resulting OC-burial rates we calculated the 'Elementary Effects' (EEs, i.e., the mean of 150 finite differences) as a measure of global sensitivity (Morris, 1991; Saltelli et al., 2008). We also computed the standard deviation of the EEs, which provides information about the degree of interactions of parameter i with the other parameters. Lastly, we calculated confidence bounds to these estimates derived via bootstrapping (SI Fig 4). For more information on the methods used and an interpretation of the results please see our updated Supplementary Methods. Motivated by the sensitivity analysis we changed porosity from a fixed value for the three depositional regimes to a spatially explicit porosity map based on (Martin et al., 2015). We have also added uncertainty envelopes (representing a $\pm 10\%$ change in the applied porosity and sedimentation rate) to our OC burial and BE estimates (Figure 3), as well as exploring the effect of individual parameters on OC burial (SI Figure 6).

SI Figure 4: Sensitivity analyses. Mean of Elementary Effects (EEs) versus their standard deviation for four key model parameters: φ representing the porosity, ω denoting the sedimentation rate, z_{bio} for the maximum depth of the bioturbated zone, and D_b for the bioturbation coefficient. Confidence bounds were derived via bootstrapping around the mean and standard deviation of the EEs. Sensitivities of OC burial rates to parameters φ , ω , z_{bio} and D_b were calculated at depths of 20, 50, 100, and 1000 cmbsf. Panel (b) shows the same as (a) but uses a different scale for the y-axis.

SI Figure 6: Burial of organic carbon in marine sediments. OC burial in marine sediments according to (a,c) sediment age, and (b,d) sediment depth. Grey shading

represents the variability in simulated OC burial arising from a change of $\pm 10\%$ in (a,b) the sedimentation rate, ω , and (c,d) porosity, ϕ .

Reviewer #3 (Remarks to the Author):

Bradley, Hülse et al., identify that the commonly used “Burial Efficiency” metric (the percentage of organic carbon hitting the seafloor that persists beyond a given depth horizon), has a number of limitations in its application and definition. In particular they use a global diagenetic model to demonstrate that depth horizons can be problematic to define burial as different sediment settings have significantly different sedimentation rates and concentrations of organic carbon. The authors suggest an updated terminology of “Transfer Efficiency” that can reflect a depth or age horizon and which better reflects the interpretation of burial. This is a concise and very well written manuscript which is to the point! The authors have identified and demonstrated some clear limitations with the use of Burial Efficiency as a metric . However, the way in which the Transfer Efficiency metric is defined and presented is not as clear as it should be. Overall, I think the manuscript is valuable and accessible to a large community.

General Comments

Definition of Transfer Efficiency - The definition of transfer efficiency towards the end of the manuscript (lines 139 - 154) led to some confusion when reading the manuscript as the term is used prior to this in Figure 3. Additionally, it took a few reads to really understand that the definition incorporated the two previous definitions of Burial Efficiency rather than a totally new metric. In part this was the language used and the fact that the definition appeared much later than the discussion of Burial Efficiency. I would suggest: 1) defining Transfer Efficiency earlier on in the text around equation 2 and before the Results and Discussion. This helps clarify the key arguments of the manuscript otherwise the transfer efficiency metric appears more like a renaming. Additionally, this directly follows the second Burial Efficiency definition and builds on that otherwise I had forgotten it was used by one paper only! ...and/or... 2) clarifying the existing definition text (see specific comments below) and updating the use of the term in Figure 3 so that the reader is not searching ahead, e.g., use two specific Burial efficiency definitions and potentially add your equivalent fully expanded transfer efficiency metric alongside.

It is certainly important that the reader understands the definition of T_{eff} and how it relates to the previous definitions of BE. We have modified the manuscript to incorporate these suggestions, including defining transfer efficiency earlier in the manuscript and using the expanded transfer efficiency metric throughout (including in the revised figures).

In general I think the new definition of transfer efficiency is useful for the reasons outlined by the authors. However, I thought it’s worth noting a perspective on the use of the term from the marine biological pump community. “Transfer Efficiency” is, as far as I’m aware, solely used to describe the POC flux reaching a depth horizon whereas a separate term, “Sequestration Depth”, is used to describe the flux reaching an age horizon (age here is defined as the average residence time of a dissolved tracer at this depth horizon) (Weber et al., 2018; Boyd et al., 2019). I don’t think this is an issue for the manuscript but could be clarified.

We have considered this point and, like the reviewer, also feel that it is not an issue for the manuscript. We favor our current approach so as to maintain the focus of the paper on OC burial in sediments, and avoid introducing a new set of metrics on the biological carbon pump in the water column that could be confusing to the reader (especially as ‘sequestration depth’

actually refers to a specific age). We ensure that we cover the timescale/length-scale relationship in Figure 2 and the corresponding text, without the need for additional terms.

Specific Comments.

Line 32: “contribute” should be “contributed”

Both tenses (‘contribute’ and ‘contributed’) work here but we favor ‘contribute’ as we describe a mechanism (independent of past, present or future), and one could also consider the present to be in a state of flux i.e. among a glacial-interglacial cycles and/or global shift in elemental cycling.

Line 52: to be asymptotic $\partial OC/\partial z = 0$ should be approximately zero or practically defined as smaller than some value?

We have modified the text and no longer refer to the asymptote (Line 62, Line 84).

Lines 72 - 74: This needs a clarifying sentence along the lines of “sediment depth represents a specific timescale of burial that is a function of local sedimentation rates”.

We have added a sentence to this effect a little earlier in the manuscript (as we feel that it is better placed here).

However, OC continues to be degraded beyond these horizons, which are often unspecified. Furthermore, different depth horizons can represent vastly different timescales of burial (largely due to differences in local sedimentation rates). The lack of clearly defined reference horizons for the calculation of BE renders this idealized notion of OC burial and preservation imprecise, inconsistent, misleading, and vague. It thus hinders the comparability of benthic OC fluxes between studies, sites and reservoirs.

(Lines 44-49)

Line 79: A reference to Figure 2a seems most appropriate.

Manuscript modified accordingly (see Lines 96-98).

Figure 2: Panel b is not described in text. The panels are largely an inverse of panel a, which makes a lot of sense, but without any description I’m not sure what i’m meant to understand from the three panels.

We have addressed this in the revised manuscript by noting the heterogeneities in the depth below the seafloor of equivalent sediment ages:

We propose that if T_{eff} 's are to be compared across settings, both depth and age should be considered. This is owing to (i) the enormous spatial heterogeneities in the age of sediment layers at fixed depths below the seafloor (Figure 2a), and similarly variable sediment depths at fixed sediment age horizons (Figure 2b), as well as (ii) the effect of changing reference depths and ages on T_{eff} (Figure 3). Studies should ideally consider both depth and time, i.e.: when specifying a reference depth, time should be discussed (and vice-versa).

(Lines 201-206)

Lines 100 - 103: is this a significant limitation or is this readily calculated? How reliable are estimates using the sedimentation rate?

We have added additional clarifying text indicating that sedimentation rates need to be complemented by other age markers (e.g. biological, chemical) to enable precise dating of sediment layers, and to mitigate and account for uncertainties linked to sediment reworking and erosion.

The limitation of this metric is that the age of a particular sediment horizon must be known or estimated (e.g., by using knowledge of past sedimentation rates, and chemical and biological age markers, whilst accounting for any post-depositional disturbances and sediment re-working).

(Lines 123-126)

Lines 117 - 118: “most sensitive to reference depths and ages...” could you briefly outline why transfer efficiency is changing most rapidly here?

We have modified the manuscript to address this point:

Values of $T_{eff,depth}$ and $T_{eff,age}$, as well as the rates of OC burial, are most sensitive to reference depths and ages in shallower (<100 cm) and younger (<10 ka) sediments (Figure 3b). These upper-most zones of sediments correspond to areas where OC degradation is fastest, due to the greater availability and preferential degradation of more reactive OC compounds (refs.^{19,20} and references therein).

(Lines 161-165)

Lines 120 - 121: is it possible to provide more specific, or at least indicative, definitions of “shallow”, “young”, and “unrealistically high”.

The precise boundaries of what constitutes too shallow or too young, as well as the impact on the specification of this limit on BE and total OC buried varies depending on the application and the question of interest. We have restructured this section, adding additional clarification on this point and also providing some indicative depths and ages for certain applications of T_{eff} .

The clear specification of reference horizons used in the calculation of $T_{eff,depth}$'s or $T_{eff,age}$'s allows for adjustments to be made to these metrics based on the characteristic (temporal or spatial) scales of the problem considered. For example, to quantify the near-instantaneous interactions between the sediment and the ocean over annual timescales, the mixed-layer depth could be specified as a depth-horizon. Alternatively, a reference depth of meters to tens of meters below the seafloor could be specified to make estimates of OC budgets on millennial to million-year timescales. What determines a suitable reference depth or age depends on the specific application and problem to be addressed. However, studies reporting BE using a reference depth that is too shallow or a reference age that is too young may convey the impression that an unrealistically high amount of OC is buried (and presumed sequestered) in sediments. This is because OC continues to be degraded beyond these horizons (in deeper and older layers) (Figure 3c).

(Lines 168-178)

Figure 3: it would help to see the abyss, margin and shelf areas delineated on a map if that's appropriate/possible?

We have added this in a new figure to the SI (SI Figure 2).

SI Figure 2. Geographic distribution of major sediment domains. Shelf (white), margin (light blue) and abyss (dark blue) domains. Shelf environments roughly correspond to water depths <200 m, with the exception of the Antarctic region where shelf area corresponds to water depths <500 m; areas deeper than ~3500 m are taken to be abyssal plain; ocean floor covered by 500 to 3500 m water are referred to as margins.

Figure 3: It would be great to use the transfer efficiency term on the figure as you have defined it, e.g., referring to specific depth or age horizons, given that this is a key point of the manuscript!

We have modified Figure 3 to incorporate this suggestion (see above).

Lines 139 - 141: “more robust metric” led me to think this was something numerically unique from the two burial efficiency definitions. I would suggest the transfer efficiency is a a new terminology that is more flexible and precise in its definition than burial efficiency.

We have modified the manuscript according to this suggestion.

We propose a new terminology, transfer efficiency (T_{eff}), for describing the fate of OC through clearly defined depth ($T_{eff,depth}$) or time ($T_{eff,age}$) horizons in marine sediments. The calculation of T_{eff} is numerically equivalent to the calculation of BE, but it requires a precise definition of spatial or temporal reference horizons.

Lines 128-131

Eqn 3: I have to admit I did not know the symbol for OR used in the equation! Could there be a quick definition in the text?)

We have changed the symbol to the more commonly used | (as in 'depth|age'), which we now feel is clear, especially since the definition of 'depth|age' immediately follows the equation.

REFERENCES

Arndt, S., Jørgensen, B. B., LaRowe, D. E., Middelburg, J. J., Pancost, R. D., and Regnier, P. (2013). Quantifying the degradation of organic matter in marine sediments: A review and synthesis. *Earth-Science Rev.* 123, 53–86. doi:DOI 10.1016/j.earscirev2013.02.008.

- Boudreau, B. P. (1997). *Diagenetic models and their implementation. Modelling transport and reactions in aquatic sediments*. doi:0.1007/978-3-642-60421-S.
- Burdige, D. J. (2007). Preservation of Organic Matter in Marine Sediments: Controls, Mechanisms, and an Imbalance in Sediment Organic Carbon Budgets? *Chem. Rev.* 107, 467–485. doi:10.1021/cr050347q.
- Dunne, J. P., Sarmiento, J. L., and Gnanadesikan, A. (2007). A synthesis of global particle export from the surface ocean and cycling through the ocean interior and on the seafloor. *Global Biogeochem. Cycles* 21. doi:10.1029/2006GB002907.
- LaRowe, D. E., Arndt, S., Bradley, J. A., Estes, E. R., Hoarfrost, A., Lang, S. Q., et al. (2020). The fate of organic carbon in marine sediments - New insights from recent data and analysis. *Earth-Science Rev.* 204, 103146. doi:https://doi.org/10.1016/j.earscirev.2020.103146.
- Martin, K. M., Wood, W. T., and Becker, J. J. (2015). A global prediction of seafloor sediment porosity using machine learning. *Geophys. Res. Lett.* 42. doi:10.1002/2015GL065279.
- Morris, M. D. (1991). Factorial sampling plans for preliminary computational experiments. *Technometrics* 33. doi:10.1080/00401706.1991.10484804.
- Muller-Karger, F. E., Varela, R., Thunell, R., Luerssen, R., Hu, C., and Walsh, J. J. (2005). The importance of continental margins in the global carbon cycle. *Geophys. Res. Lett.* 32. doi:10.1029/2004GL021346.
- Saltelli, A., Ratto, M., Andres, T., Campolongo, F., Cariboni, J., Gatelli, D., et al. (2008). *Global sensitivity analysis: The primer*. doi:10.1002/9780470725184.
- Ullman, W. J., and Aller, R. C. (1982). Diffusion coefficients in nearshore marine sediments. *Limnol. Oceanogr.* 27. doi:10.4319/lo.1982.27.3.0552.

REVIEWERS' COMMENTS

Reviewer #2 (Remarks to the Author):

I think the reviews have done a thorough and excellent job of addressing all previous comments and concerns. I recommend this article be published as is.